# Evaluation of Rapid Dot-Immunoassay for Detection Orthopoxviruses Using Laboratory-Grown Viruses and Animal’s Clinical Specimens

**DOI:** 10.3390/v14112580

**Published:** 2022-11-21

**Authors:** Nikita Ushkalenko, Anna Ersh, Alexander Sergeev, Pavel Filatov, Alexander Poltavchenko

**Affiliations:** State Research Center of Virology and Biotechnology “Vector”, Novosibirsk Region, Koltsovo 630559, Russia

**Keywords:** orthopoxviruses, monkeypox virus, cowpox virus, vaccinia virus, ectromelia virus, rabbitpox virus, detection, clinical samples, dot-immunoassay

## Abstract

The aim of the work was an experimental evaluation of the characteristics of the kit for the rapid immunochemical detection of orthopoxviruses (OPV). The kit is based on the method of one-stage dot-immunoassay on flat protein arrays using gold conjugates and a silver developer. Rabbit polyclonal antibodies against the vaccinia virus were used as capture and detection reagents. The sensitivity of detection of OPV and the specificity of the analysis were assessed using culture crude preparations (monkeypox virus, vaccinia virus, rabbitpox virus, cowpox virus, and ectromelia virus), a suspension from a crust from a human vaccination site as well as blood and tissue suspensions of infected rabbits. It has been shown that the assay using the kit makes it possible to detect OPV within 36 min at a temperature of 18–40 °C in unpurified culture samples of the virus and clinical samples in the range of 10^3^–10^4^ PFU/mL. Tests of the kit did not reveal cross-reactivity with uninfected cell cultures and viral pathogens of exanthematous infections (measles, rubella and chicken pox). The kit can be used to detect or exclude the presence of a virus threat in samples and can be useful in various aspects of biosecurity. The simplicity of analysis, the possibility of visual accounting the and interpretation of the results make it possible to use the test in laboratories with a high level of biological protection and in out-of-laboratory conditions.

## 1. Introduction

As a result of the successful eradication of smallpox, mass vaccination against it was discontinued in 1980. At the same time, the possibility of using the variola virus (VAV) for terrorist purposes [1,2], as well as diseases caused by the virus monkeypox (MPXV), and other orthopoxviruses (OPV) remain a threat to the unvaccinated population [3,4,5], especially in MPXV endemic regions of Central and West Africa. These are mostly rural areas with limited resources [4,6]. Recently, the distribution of MPXV beyond endemic areas has become a serious threat. The latest cases of monkeypox spread outside of Africa have become widespread since May 2022 and have been reported in the USA, UK, Australia, Belgium, Canada, Georgia, South Korea and other countries. Since the start of the monkeypox outbreak and as of 30 September 2022, 68,428 laboratory-confirmed cases have been reported to WHO from 106 countries/territories in five WHO Regions [7]. So far, only two cases of MPXV infection has been registered in Russia, but the likelihood of a larger importation of the virus is quite real.

The diagnosis of OPV infection cannot be made on clinical grounds alone, since similar symptoms are characteristic of a number of other exanthematous diseases [8]. Differential diagnosis can be made by laboratory methods, but rapid and reliable detection of infectious agents in the absence of well-equipped laboratory infrastructure is a difficult task. For such areas, methods based on PCR and ELISA, which require qualified specialists, expensive equipment and strictly controlled laboratory conditions, are not suitable [9]. Quick and easy diagnostic methods could be useful here, but Point-of-Care (POC) methods for detecting OPV are limited and need to be improved. For example, the lateral flow immunoassay Orthopox BioThreat^®^ Alert can be performed within 20–25 min, but is capable of detecting OPV at a concentration of 10^6^–10^7^ PFU/mL [9]. Another immunofiltration system, ABICAP (Antibody Immuno Column for Analytical Processes), makes it possible to detect OPV at their concentration in the sample exceeding 10^4^ PFU/mL, but it is performed within 45 min [10]. The known maximum sensitivity of the immunochemical detection of OPV (10^3^ PFU/mL) was achieved in ELISA using carefully selected monoclonal antibodies [11].

In a previous publication [12], we reported on the promise of using a fast and easy-to-perform dot immunoassay on flat protein arrays to detect OPV. This publication reports the results of laboratory testing of this test using a wide range of orthopoxviruses, including human pathogenic monkeypox viruses, as well as clinical materials from infected laboratory animals.

## 2. Materials and Methods

### 2.1. Materials

In our work we used: bovine serum albumin, casein from bovine milk, Gold (III) chloride hydrate, peptone, sodium azide, sucrose and Tween-20 (Sigma, Burlington, MA, USA); Russian-made chemical reagents of analytical purity grade. IgG from rabbit hyperimmune serum (a/POX-IgG) and IgG from normal rabbit swas (NRS-IgG) were isolated by precipitation with ammonium sulfate.

Synthetic Paper «Pentaprint» of PR-M480/09-07/8101-2D8 brand (Klöckner Pentaplast, Montabaur, Germany) was used for the manufacture of the substrate for protein. The analytical baths were sealed using the «Coflex» covering material (Firm Sigma, Russia).

### 2.2. Viruses

Monkeypox virus, strain V79-1-005 (MPXV) [13], vaccinia virus strain LIVP (VACV) [14], vaccinia virus recombinant strain ABCNJ (VACV/ABCNJ) [14], vaccinia virus with amino acid substitutions D110N, K151E in membrane glycoprotein A34 (VACV_A34R_[D110N_K151E]) [15], cowpox virus, strain GRI-90 (CPXV) [16], ectromelia virus, strain K-1 (ECTV) [17], and rabbitpox virus, strain Uetrecht (RPXV) were used in this study.

Propagation and titration of viruses on cell culture CV-1 were conducted according to previously described procedures [12]. All viruses were used natively as clarified supernatant of infected cells cryolisates. All work with live MPXV was conducted within a biosafety level 4 laboratory.

#### 2.2.1. Antibodies

a/POX-IgG—IgG from rabbit hyperimmune serum. Hyperimmune sera against total viral proteins were obtained by immunizing chinchilla rabbits with a purified HDL strain of the vaccinia virus. Animals were immunized three times with an interval of 28 days. The first immunization was carried out at a dose of 8.2 × 10^6^ PFU/animal, intradermally in two places. The second immunization was carried out at a dose of 1.2 × 10^8^ PFU/animal using complete Freund’s adjuvant in a ratio of 1:1 subcutaneously in 10 places. The third immunization was carried out similarly to the second, but using incomplete Freund’s adjuvant. Blood was collected 28 days after the third immunization and serum was separated by fibrin clot formation and retraction. IgG from rabbit immune serum (a/POX-IgG) was isolated by precipitation with ammonium sulfate and stored at −18 °C.

IgG NRS—IgG from normal rabbit serum, isolated by precipitation with ammonium sulfate, stored at −18 °C.

#### 2.2.2. Clinical Materials

The detection of OPV in blood, nasal mucosal swabs, and skin lesions was assessed on materials obtained from three 6-month-old chinchilla rabbits weighing 2.0–2.5 kg, intranasally infected with RPXV at a dose of 10^4^ PFU/animal.

Animal studies and manipulations were approved by the Bioethics Committee of the FBSI SRC VB “Vector” of Rospotrebnadzor and were carried out in accordance with veterinary legislation and in accordance with the requirements for the humane maintenance and use of animals in experimental studies [18].

The fallen-off crust from the place of vaccination of the patient was obtained from medical unit №163 in the form of an anonymous sample.

#### 2.2.3. Colloidal Gold Conjugate Au-a/POX-IgG

A gold sol (15–20 nm) was obtained by reducing a solution of tetrachloroauric acid with sodium citrate at 100°C. The coagulation test to determine the dose of the sol loading with a/POX-IgG antibodies and the loading itself was performed as described previously [19]. Additional stabilization of the sol was performed by adding BSA to 1%. The conjugate was purified by centrifugation at 14,000× *g* for 35 min at 4 °C, 40% glycerol was added and stored at −20 °C.

#### 2.2.4. OPV Detection Kit

The OPV detection kit included protein arrays, analytical baths, a perforator for opening the foil over the bath wells, vials with double-distilled water and 0.4% silver nitrate solution. The view of the elements of the kit is shown in Figure 1A. 

Protocols for the manufacture of protein arrays and the preparation of analytical baths are given in a previous work [12]. Briefly: in the preparation of protein arrays, working dilutions of capture reagents in 5 mM borate buffer solution (pH 6.0) were applied in aliquots of 2.5 μL to a synthetic paper substrate according to the scheme shown in Figure 1B, dried for 2 h at 45 °C, blocked in 0.2% casein solution, dried and used in work. The polypropylene analytical bath combines 5 modules of 9 wells, each of which is designed to perform 1 analysis. The bath wells were filled in accordance with Table 1. 

Upon receipt of the developer, 200 μL of purified water was added to the wells of the 6th row of the bath to dissolve the dry mixture tablets and, just before the development, 200 μL of 0.4% silver nitrate solution was added to the wells. An alkaline solution of thiourea in the 8th row of the bath was used to enhance and stabilize staining.

#### 2.2.5. Dot-Immunoassay

The analysis was carried out at a temperature of 20–25 °C. The scheme of dot-assay is shown in Figure 2.

The samples in a volume of 100 μL were introduced into wells (row 1) with the conjugate, protein arrays were placed in the resulting mixture and incubated for 25 min, then the arrays were moved along the next rows of wells to perform washing and development procedures. Total analysis time 36 min. To assess the sensitivity of the assay, a series of two-fold dilutions of viral suspensions were used. The limit of detection of viruses was calculated as the result of multiplying the initial titer of the virus sample by the maximum dilution at which the test point was clearly defined visually. Specificity was assessed using preparations of non-infected cell culture and with heterogeneous controls of the causative agents of exanthematous infections: Measles, Rubella and Chickenpox viruses. All experiments were carried out in 3 repetitions and showed the same limit of detection. Since the measurements were not quantitative, statistical processing of the results was not carried out.

## 3. Results

Observation of infected rabbits was carried out for 7 days from the moment of infection to the death of animals. After 4 days, the onset of the disease was noted in rabbits, which was accompanied by an increase in temperature. Visible signs of infection, such as tremors and rhinitis, began to be noted after 6 days, and after 7 days a maculopapular skin rash appeared. During the entire infectious period, 3 mL of blood was taken daily from the ear vein, and after the formation of a clot, each sample was divided into serum and formed elements. After 6 days, secretions from the nasal mucosa were collected with a cotton swab moistened with physiological saline (PS), the swab was squeezed several times in 1 mL of PS, and the resulting liquid was used for analysis. After the death of the rabbit, pieces (0.5 cm^2^) of skin were cut out from the area of visible rashes. Ten percent homogenates in PS were prepared from a clot of blood cells and selected skin samples. The resulting homogenates in a volume of 100 µL were added to the wells of the 1st row of the analytical bath and a dot analysis was performed.

The results of a comparative study of serial dilutions of the VACV (LIVP) preparation and a suspension from a crust from a pustule at the site of human vaccination, which fell off on the 29th day after the introduction of vaccinia, are shown in Figure 3.

The results of RPXV detection in tissue samples from infected rabbits are shown in Figure 4.

The results of estimates of the sensitivity of rapid dot-immunoassay for various laboratory-grown orthopoxviruses are given in Table 2.

## 4. Discussion

With the eradication of smallpox in 1980 and the subsequent cessation of smallpox vaccination, monkeypox has emerged as the most important OPV for public health. Monkeypox is especially dangerous in endemic areas covered by rainforests in Central and West Africa. These are mostly rural regions with limited resources and insufficient diagnostic infrastructure [4,6]. Limited diagnostic capacity does not allow complex tests such as PCR or ELISA, which require qualified specialists, expensive equipment and strictly controlled laboratory conditions [9]. For such areas, simple methods that can be performed in the field are necessary.

In a previous work [12], we compared the efficiency of OPV detection in routine (two-stage) and rapid (one-stage) dot analysis. Reduction of the execution time of the fast version of the analysis is achieved by combining the stages of incubation of protein arrays in the sample and conjugate and reducing the number of washes. In contrast to the previously described variants of immunochemical analysis [9,10,11], which use pairs of monoclonal antibodies against different antigenic determinants of OPV, our method is based on the use of polyclonal antibodies against VACV, both as a capture reagent immobilized on a substrate and as bound with particles of colloidal gold detection antibodies. Previously, it was established that the limit for the determination of VACV has an inverse relationship with the degree of purification of preparations from subviral structures; and the single-stage variant of dot-immunoassay, along with a reduction in the analysis time, makes it possible to increase the sensitivity of detecting VACV in crude viral preparations. It is shown that such an increase in sensitivity can be explained by the formation of large aggregates of gold conjugate particles on the surface of subviral structures, which significantly enhance the optical signal when the analysis results are displayed.

In the present work, we have extended this study using a wide range of crude OPV preparations, including the human pathogen MPXV, as well as clinical specimens from humans and infected animals.

Monkeypox can clinically resemble various rash illnesses which need to be considered during differential diagnosis. It includes measles, rubella, chickenpox and some other infections. Chickenpox is most commonly confused with monkeypox (up to 50% of cases in some outbreaks) [20] due to the similarity in the clinical presentation of the two diseases. Dot analysis is specific and does not detect interactions with heterogeneous controls of pathogens of exanthematous infections (measles, rubella and chicken pox viruses) and with uninfected cell culture preparations.

MPXV-infected patients are contagious from the onset of fever to the separation of crusts and the formation of a fresh layer of skin. The virus is present in the blood and mucous membranes during the fever and early rash stages, as well as in skin lesions at all stages of the disease. For field research, the most suitable samples are blood in the fever stage, although some authors do not recommend blood sampling for diagnostic purposes, since the timing and duration of viremia are different, and the results are often inconclusive [21]. As samples, nasal mucosal swabs, as well as liquid or crust at the rash stage can be considered.

Only 2 cases of monkeypox were reported in Russia and clinical materials from this patient were not available. Explored the crust from the pustule (weighing 0.2 g) at the site of vaccination, which fell off on the 29th day after the introduction of cowpox. The dried crust was ground in a porcelain mortar and resuspended in 1 mL of physiological saline (PS). An approximate assessment of the concentration of VACV in the suspension from the crust was performed by parallel titration of this suspension and the VACV (LIVP, 8.5 × 10^6^ PFU/mL) preparation. A series of dilutions in PS was prepared and a dot-immunoassay analysis was performed. The results of the evaluation are shown in Figure 3.

Figure 3 shows that in terms of the colour intensity of the spots in the test area and the maximum dilution in which the samples give a positive optical signal, the preparation from the crust lags behind the viral material by an order of magnitude. Therefore, the concentration of the virus in the suspension from the crust is approximately 10 times less than in the original VOV(LIVP) preparation and is about 10^6^ PFU/mL, and the detection limit, respectively, is about 5 × 10^3^ PFU/mL.

To assess the detection of OPV in the blood, nasal mucosal swabs and skin lesions, three 6-month-old chinchilla rabbits weighing 2.0–2.5 kg were used intranasally. infected with the RPXV virus at a dose of 10^4^ PFU/animal.

Figure 4A shows that the virus is detected in the blood serum of infected animals from the 4th day after infection, which coincides with an increase in body temperature, and in the following days its content increases sharply until the death of rabbits. Very high titers of the virus in the blood on the 5th and 6th days are indicated by a decrease in the colour intensity of the C+ zone on the matrix (the so-called “hook effect”), indicating that the conjugate in the analysis mainly binds to viruses in the liquid phase, which leads to a deficiency of the gold conjugate that reacts with the antigen at the control point on the surface of the matrix. We have previously noted this effect in the analysis of OPV with a titer of more than 10^5^ PFU/mL (unpublished data). In blood cells, virus antigens are detected a day later (see Figure 4B). All rabbits showed similar results.

Figure 4C shows that nasal discharge and skin rash (papules) in all rabbits contain approximately the same high concentrations of the virus. The results obtained are consistent with previously published data [22].

The results of rapid dot immunoassay sensitivity assessments for various laboratory-grown orthopoxviruses are shown in Table 2. Table 2 demonstrate that the minimum detection limit was achieved in the RPXV assay of 6.2 × 10^2^ PFU/mL and the maximum detection limit was achieved in the CPXV assay of 8.0 × 10^3^ PFU/mL Taking into account possible errors in virus titration and dot-analysis, the sensitivity of the rapid version of the detection of OPV in crude preparations can be designated in the range of 10^3^–10^4^ PFU/mL. This sensitivity is slightly below the detection limit for ELISA using carefully selected pairs of monoclonal antibodies [11], but exceeds the sensitivity or speed of known POC systems for outside laboratory diagnostics of OPV (the lateral flow immunoassay Orthopox BioThreat^®^ Alert can be performed within 20–25 min, but is capable of detecting OPV at a concentration of 10^6^–10^7^ PFU/mL [9] and immunofiltration system, ABICAP, makes it possible to detect OPV at their concentration in the sample exceeding 10^4^ PFU/mL, but it is performed within 45 min [10]). Dot analysis is specific (during the execution of all studies, false positive results were not registered) and does not detect interactions with uninfected cell culture preparations and with heterogeneous controls of pathogens of exanthematous infections (measles, rubella and chicken pox viruses).

The presented method is simple to perform and does not require equipment and a power supply, which makes it suitable for use directly in the focus of infection (at the patient’s bedside) or under the conditions of using protective equipment in a specialized laboratory. This approach avoids the need for sample inactivation measures, which may adversely affect the sensitivity of OPV detection.

The large central region of OPV genome is highly conserved which explains the significant degree of crossreactivity of different OPV in immunochemical tests [23]. The described dot-immunoassay also allows for the identification of OPV at the genus level only. However, in combination with characteristic symptoms, this method makes it possible to establish infection with pathogenic types of poxviruses, which is sufficient to take urgent measures to isolate and treat patients. In addition, the dot immunoassay is robust and practical to out of the laboratory. Confirmation of the MPXV infection is best conducted by polymerase chain reaction (PCR) as it is the only method which can differentiate between the orthopoxvirus species [24,25].

## 5. Conclusions

The dot immunoassay described above is a sensitive, specific, rapid, easy-to-perform, and inexpensive test that detects OPV in crude virus samples and most available clinical samples in the range of 10^3^–10^4^ PFU/mL within 35 min. A significant advantage of the test described above is that it is produced using the same type of polyclonal antibodies both as a capture reagent and as a detection reagent, which greatly simplifies and reduces the cost of manufacturing a diagnostic system [26]. Although only a few strains of OPV were tested in the present study, they are known to have broad antigenic cross-reactivity [23] and results should be similar for all OPVs, including the highly pathogenic variola virus. This method of analysis is applicable for studies in accordance with the BSL 3 biocontrol protocol and can be useful in detecting a virus threat in various biosecurity applications.

## 6. Patents

As a result of this work, a patent RU 2 729 635 was obtained, priority dated 27 May 2019.

## Figures and Tables

**Figure 1 viruses-14-02580-f001:**
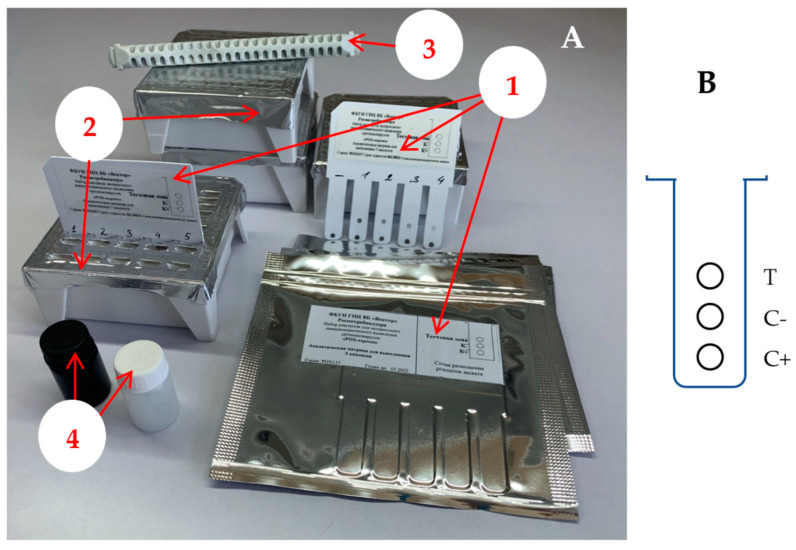
Design of a kit for dot-immunoassay of orthopoxviruses. (the **A**)-the basic elements of the kit for detection of orthopoxviruses: 1-protein arrays, 2-analytical baths, 3-perforator, 4-liquid components of the developing system. (**B**)-the scheme of capture reagents immobilization on the protein array: T-test zone (a/POX-IgG 1:100), C−-negative control (IgG RNS 1:100) and C+-positive control (VACV (LIVP) 10^5^ PFU/mL).

**Figure 2 viruses-14-02580-f002:**
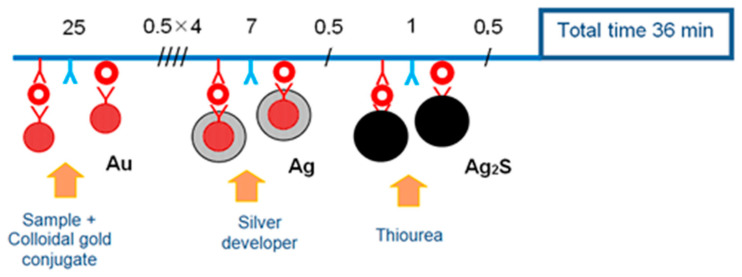
The scheme of the rapid dot-immunoassay of orthopoxvirus using conjugate of colloidal gold with polyclonal antibodies; enhanced optical signal by silver developer and stabilized staining with alkaline solution of thiourea. The top lines show the multiplicity and duration (min) of operations. 
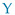
—antibodies,
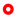
—antigens,
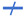
—the multiplicity of washing.

**Figure 3 viruses-14-02580-f003:**
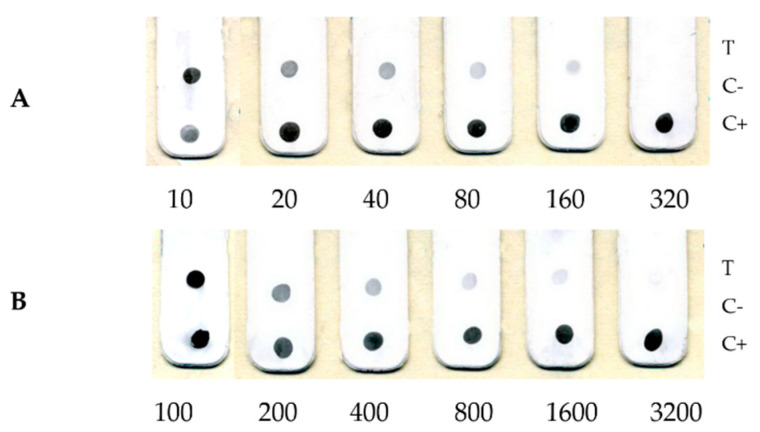
Comparative assay of the dilution series: (**A**)-suspension from the crust from the pustule at the site of vaccination; (**B**)-VACV(LIVP) preparation (initial titer 8.5 × 10^6^ PFU/mL). The numbers below indicate the multiplicity of dilutions.

**Figure 4 viruses-14-02580-f004:**
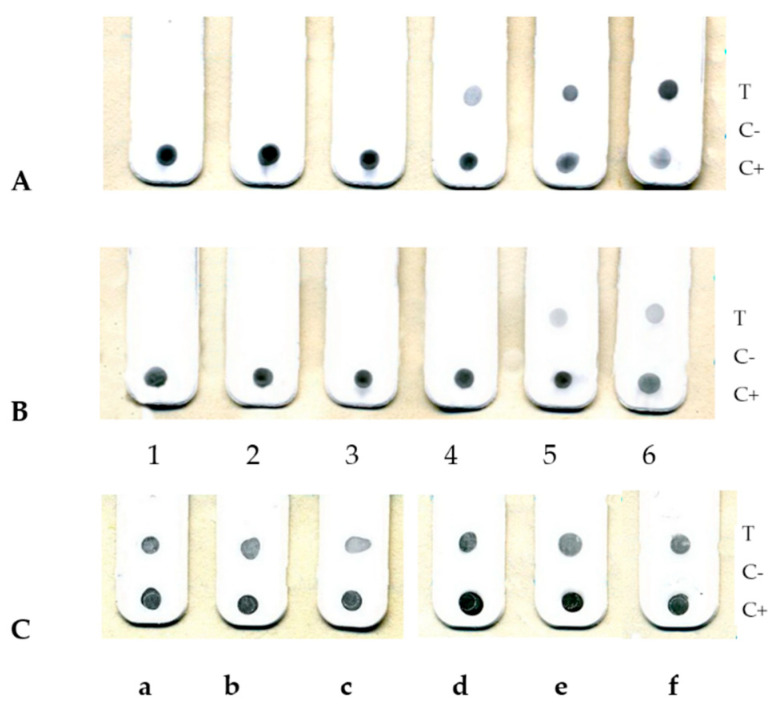
The results of dot-immunoassays of clinical samples from an infected rabbit. Type of protein arrays after the assays: (**A**)-blood serum and (**B**)-blood cells (the numbers under the arrays indicate the days from the moment of infection); (**C**)-washes from the nasal mucosa (**a**–**c**) and homogenates of samples skin areas with papular rashes (**d**–**f**) every rabbit.

**Table 1 viruses-14-02580-t001:** Analytical bath wells content.

Row	Composition of the Working Solution	Volume, µL
1	dilution of gold conjugate in PBS-T with 0.02 % PEG-20000, pH 7.4	200
2	PBS-T	300
3	PBS-T	350
4	purified water	400
5	purified water	400
6	tablet of a dry mixture of citric acid and metol in a ratio of 5:3	4 mg
7	purified water	400
8	1% thiourea and 1% sodium hydroxide in purified water	300
9	purified water	400

Note: PBS-T—0.02 M sodium phosphate buffer solution with 0.8 % NaCl, 0.1 % Tween-20 and 0.1 % sodium azide, pH 7.2 in purified water.

**Table 2 viruses-14-02580-t002:** Sensitivity of orthopoxviruses detecting by dot-immunoassay.

Sample		Initial Titer, PFU/mL	Detection Limit of Dot-Immunoassay	
			Limiting Dilution	Titer, PFU/mL
Monkeypox virus, strain V79-1-005		4.0 × 10^6^	1/1600	2.5 × 10^3^
Vaccinia virus, strain LIVP		8.5 × 10^6^	1/12,800	6.6 × 10^2^
Vaccinia virus, strain ABCNJ		1.3 × 10^7^	1/3200	4.1 × 10^3^
Vaccinia virus, _A34R_ [D110N_K151E]		1.1 × 10^6^	1/800	1.4 × 10^3^
Ectromelia virus, strain K-1		8.5 × 10^6^	1/400	5.6 × 10^3^
Cowpox virus, strain GRI-90		1.3 × 10^7^	1/1600	8.0 × 10^3^
Rabbitpox virus, strain Uetrecht		1.0 × 10^6^	1/1600	6.2 × 10^2^
Cell culture control		0	0	0
Heterogenous controls	Chickenpox virus	0	0	0
	Measles virus	0	0	0
	Rubella virus	0	0	0

## Data Availability

Not applicable.

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
