# Peer review of "Evaluation of Rapid Dot-Immunoassay for Detection Orthopoxviruses Using Laboratory-Grown Viruses and Animal’s Clinical Specimens"

_viruses, 2022, doi:10.3390/v14112580_

Round 1
Reviewer 1 Report (Previous Reviewer 1)
The revised MS provided sufficient detail in this version. One editorial revision is recommended below:
1. Line 19-20, the sensitivity range is often described from low to high end. Please follow the tradition and revise as 103 -104 PFU/ml
Author Response
Dear Colleague,
Thank you for your valuable advice, which allowed us to improve the article. We have corrected the designation to 103 -104 PFU/mL
Reviewer 2 Report (Previous Reviewer 2)
I appreciate the author's efforts to revise this manuscript according to the reviewer's suggestion and believe that it significantly improved. There is a minor issue, which the authors might consider before publication:
Lanes 98-113 describe in great detail infection, pathogenesis, and concomitant read-out of the do-blot analyses. This paragraph, to my opinion, would rather fit to the "result" section, while keeping a short description of the method (initial infection titer; sampling method and intervals; analyses).
Author Response
Dear Colleague,
Thank you for your valuable advice, which allowed us to improve the article. We agree with the comment and edited the paragraph accordingly. Brief information about biomaterials from rabbits is presented in the Materials and Methods section. We have moved detailed information about the sample preparation process to the “Results” section.
This manuscript is a resubmission of an earlier submission. The following is a list of the peer review reports and author responses from that submission.
Round 1
Reviewer 1 Report
The aim of the work was to evaluate a patented rapid immunochemical detection kit for orthopoxviruses. A fast detection kit to confirm the presence of orthopoxviruses in culture or clinical specimens is needed in the field at this time. Thus, the work has applicable importance during the ongoing monkeypox virus outbreak for diagnosis of biospecimens in various situations. However, the manuscript will benefit from a restructure by providing sufficient information in the material and method and the result sections and by including appropriate discussion in the “Discussion” section. Currently the “Discussion” section contains content that is mostly fitting in the result section. Below are suggestions to improve the MS:
1. The important question is whether the assay can reliably process the clinical samples consistently. Thus, to set up the standard, important details of information such as the buffer conditions in which the clinical samples are resuspended or processed must be included in the material and method section. If alternative buffer conditions can also, it will be helpful to include them as well.
2. In the experiment result section for Table 1, Table 2, Figure 3-5, there is no description on how many times the biological replicates and technical replicates are performed resulting in the conclusion on detection sensitivity. For rabbit tissue test, the specimens were collected from one rabbit infected with a lethal dose of Rabbitpox virus. It is not clear how many BALB/c mice were used per dose of 10^6-10^8 pfu/30 uL. In either case, these are high lethal doses of infection. It is important to evaluate the detection limit from animals representing realistic clinical situations. It will be beneficial for its future clinical application to show the ability of this immunoassay for detecting orthopoxviruses from animal specimens with sub-lethal doses of orthopoxvirus infections as well. It is possible that at an early time point during the infection, detection can be successful in a tissue dependent manner and it will be informative to present the data.
3. In Figure 3, it needs to be clearly labeled what are the dot results represent in at least one of 3B, 3C, and 3D. It is speculated that the dot at the bottom of each strip is the positive control, but it is not clear where is the negative control on the actual strip. Is it true none of strips show any false positive in the negative control, and in Fig 3-5 among all the strips the dots above the positive controls are from test samples?
4. Consistent with the question from “1”, in Table 2 how these specimens were obtained (technical details) and what buffer was used for the initial sample preservation or resuspension will need to be provided. How are the specimens divided for titration vs. dot immunoassay? Some information is provided in the discussion (line 239-250) and appropriate additional detail will be needed in the material and method and the result sections.
5. In Figure 4, detection of virus in nasal septum with mucosa from 10^6 pfu inoculated animal showed no positive signal, which showed 4x10^3 pfu/mL viral titer. The positive detection of the nasal septum specimens from 10^8 pfu inoculated animal showed 10^7 pfu/mL titer. What is the detection limit for nasal septum with mucosa using viral titer as a standard? Realistically, nasal septum and blood will most likely be the specimens for clinical diagnosis in human subjects. Additional information will be needed to evaluate the clinical value of this potential diagnosis tool to strengthen its use in the clinical setting.
Reviewer 2 Report
The current manuscript describes a new method for the detection various orthopox virus infections, including monkey pox, on the basis of immune dot blots. The already patented method is indeed of interest, since it does not require expensive equipment and thus might become suitable in less accessible areas around the world, in which orthopox virus infections are still a considerable threat.
Most importantly, the presented results appear sound, are well controlled and both, sensitivity and specificity appear to be appropriate. However, there are some concerns about the structure and the presentation of the manuscript, which should be addressed prior to publication:
1) In particular, the Discussion rather presents results than interpretations and potential future impact. For instance it could be of potential interest to mention perspectives regarding the discrimination of OPV species, since the presented method just addresses the general detection of all kind of OPVs.
2) Figure 1 and 2 are not really informative and should contain more details. It would indeed make sense to indicate the composition of the components and a detailed application protocol.
3) In lane 150ff the authors indicate that the experiments were carried out with several repetitions – how many?
4) All results presented in this manuscript are solely based on samples, which were derived from controlled inoculated species. However, it would be much more convincing, if the authors could additionally present some screening results.
5) Figure 3: The position indicated in Fig. 3A should be indicated on the level of the strips. This would facilitate easier assessments of the results.